# The Healing-Promoting Properties of Selected Cyclitols—A Review

**DOI:** 10.3390/nu10121891

**Published:** 2018-12-03

**Authors:** Agnieszka Owczarczyk-Saczonek, Lesław Bernard Lahuta, Magdalena Ligor, Waldemar Placek, Ryszard Józef Górecki, Bogusław Buszewski

**Affiliations:** 1Department of Dermatology, Sexually Transmitted Diseases and Clinical Immunology, University of Warmia and Mazury in Olsztyn, 10-229 Olsztyn, Poland; w.placek@wp.pl; 2Department of Plant Physiology, Genetics and Biotechnology, University of Warmia and Mazury in Olsztyn, 10-229 Olsztyn, Poland; lahuta@uwm.edu.pl (L.B.L.); rigorecki@gmail.com (R.J.G.); 3Department of Environmental Chemistry and Bioanalytics, Faculty of Chemistry, Nicolaus Copernicus University, 87-100 Torun, Poland; magdalena.ligor@umk.pl (M.L.); bbusz@umk.pl (B.B.)

**Keywords:** cyclitols, d-*chiro*-inositol, *myo*-inositol, d-pinitol, insulin resistance, diabetes, polycystic ovarian syndrome

## Abstract

Introduction: *Myo*-inositol and its derivatives cyclitols play an important role in the processes of cell regulation, signal transduction, osmoregulation, and ion channel physiology, and are a component of the cell membrane. Free cyclitols present in food or released during the degradation of galactosyl cyclitols by bacteria (in digestive tract) show some physiological benefits. Aim: The aim of this paper is to present and analyze the documented data about curative and healing properties of cyclitols. Results and discussion: Cyclitols are well known compounds in the treatment of an accompanied diabetes insulin resistance, and also obesity and polycystic ovarian syndrome. d-*chiro*-Inositol deficiency exacerbates insulin resistance in the liver, muscles, and fat, while depletion of *myo*-inositol results in the development of diabetic complications. Cyclitols are successfully applied in treatment of polycystic ovarian syndrome, simultaneous are observed effective reducing of BMI, improving the hormonal profile, and increasing fertility. Moreover, cyclitols have anti-atherogenic, anti-oxidative, anti-inflammatory, and anti-cancer properties. Conclusion: The properties of cyclitols may be a good therapeutic option in the reduction of metabolically induced inflammation. Due to well drugs tolerance and low toxicity of these compounds, cyclitols are recommend for pregnant women and also for children. Another advantage is their widespread presence and easy availability, which encourages their use in medicine.

## 1. Background

*Myo*-inositol (MI) and its derivatives Inositols (hexahydroxycyclohexanes) are widely distributed in the tissues without coma, of mammals, plants, fungi, and some bacteria. They play an important role in the processes of cell regulation, signal transduction, osmoregulation, ion channel physiology, and are a component of the cell membrane [1,2]. Inositol is found in all living organisms and is synthesized from glucose-6-phosphate by *myo*-inositol-3-phosphate synthase to form *myo*-inositol-3-phosphate (Ins3P), which is dephosphorylated to MI [3]. In plants, inositol hexaphosphate is in a form of phytic acid or its salt, as a phosphorus reserve [4]. In many cases, molecular similarity to sugars negatively influences the bio-processes in cellular level by interferences. It is difficult to obtain their pure form. Therefore, bioanalytics plays a very important role in the course of metabolism of these compounds [1,2]. In terms of chemical structure, they are cyclitols (cycloalkanes), because they contain a hydroxyl group on each of three or more ring atoms (Figure 1A).

*Myo*-inositol is only one of nine possible structural isomers of inositol (1,2,3,4,5,6-cyclohexanehexol). Among cyclitols, well-known *myo*-inositol is widely present in nature. There are nine possible isomers of *myo*-inositol. Five of them naturally occur like *myo*-, *scyllo*-, *muco*-, and *neo*- and D-*chiro*-inositol, while other four isomers (L-*chiro*, *allo*-, *epi*-, and *cis*-inositol) are derived from *myo*-inositol and obtained by chemical synthesis [1,5]. However, the number of possible inositol phosphates of those nine isomers (excluding pyrophosphates) is 357 [6]. Some methyl derivatives of inositols are secondary plant metabolites that do not normally exist, but are produced only in response to abiotic stress, such as salinity, cold or high temperature [1,7,8,9]. Among them, we can distinguished: 5-*O*-methyl-*myo*-inositol (sequoyitol), 1-*O*-methyl-*myo*-inositol (bornesitol), 4-*O*-methyl-*myo*-inositol (ononitol), 5-*O*-methyl-*allo*-inositol (brahol), di-*O*-methyl-(+)-chili-inositol (pinpollitol), 3-*O*-methyl-d-*chiro*-inositol (d-pinitol-DP) and 1L-2-*O*-methyl-*chiro*-inositol (L-quebrachitol) [1]. Cyclitols are biosynthetically derived from glucose via Loewus pathway to give the free *myo*-inositol which can be converted via methylation to give the other cyclitols (Figure 2) [2,10].

Inositol was classified as the group of B vitamins, but it can be synthesized in the human body from phytic acid by a phytase enzyme [11]. In human and mammalian cells, 99% of inositol is in the form of MI, and only 1% exists in the form of DCI [4]. The daily diet provides about 1 g of inositol, mainly in the form of MI and which is sufficient to cover the requirement for this substance. Its natural sources are: citrus fruits, whole grains, nuts, wheat germs, legumes, and yeast [11,12]. Exogenous MI is absorbed in the intestines and then it is deposited in the brain, heart myocardium, and skeletal muscles, as well as in the bones and gonads. It is also produced by saprophytic bacteria [11,12,13]. *Myo*-inositol is the precursor of inositol triphosphate, a second messenger generated after the activation of phospholipase-C and regulating many hormones such as thyroid stimulating hormone (TSH), follicle stimulating hormone (FSH), and insulin [14,15].

MI and other cyclitols indicate several health-promoting and even healing properties [13]. Many studies confirm their beneficial effects on the metabolism of animals and humans (Table 1 and Table 2). Insulinomimetic effects of d-pinitol (DP), d-*chiro*-inositol (DCI), and sequoyitol are very well documented [16,17,18]. MI can improve osteogenesis and bone mineral density with DCI, suppressesing osteoclastogenesis [19,20]. MI additionally has antioxidant [21], anti-inflammatory [22], and even anti-cancer properties [23,24,25].

Cyclitols, as active compounds, can affect enzymes involved in the drug’s metabolism. It has been proven that DP, originating from the African plant *Sutherlandia frutescens* (Wildegansie), reduces the accumulation of atazanavir (used to treat HIV/AIDS) in intestinal epithelium microsomes and inhibits its metabolism in hepatocyte microsomes. The result of these processes is to increase the bioavailability of the drug and its longer duration of half-time [42].

As non-toxic compounds, cyclitols extensive occurrence in nature, and availability of mentioned compounds, encourages their wider use in therapy (Table 3).

## 2. Cyclitols, Insulin Resistance, and Diabetes

Diabetes mellitus affects up to 422 million adults and has a steady upward trend, mainly due to the obesity epidemic in highly developed countries. Diabetes and complications associated with this disease are the fourth leading cause of mortality all around the world. It leads to cardiovascular diseases, blindness, and kidney failure. This disease is a major economic burden for health care, therefore researches of cyclitols using in treatment may have very important practical implications.

The studies proved the beneficial effect of DCI and its derivatives (l-quebrachitol, d-pinitol, pinpollitol, and d-ononitol), which have an effect similar to insulin, causing a reduction in blood glucose. Until now, the effects of cyclitols from the following plants have been proven in clinical trials: soybean, buckwheat, carob tree, fig, mung bean, and fig tree (used in Asia as an antipsoriatic agent).

Insulin also works by means of mediators that consist of inositols linked to hexosamines, hexoses, organic phosphates and ethanolamines, and are referred to as inositol phosphoglycans (IPG) [12,43]. They mimic the effects of insulin in vivo. IPG, which contains MI inhibits purified AMP-dependent cyclic protein kinase, and IPG containing DCI, activates pyruvate dehydrogenase phosphatase [12,43]. In diabetic patients, the concentration of DCI and bioactivity in tissues is lowered, whereas insulin administration increases [43,44]. This phenomenon is also accompanied by a significant reduction in muscles’ DCI- biological activity in people with type 2 diabetes compared to the control group [43,44]. In addition, the bioactivity of IPG-DCI in diabetic patients increases up to 10-fold, with a significant decrease in plasma MI after insulin injections [43]. Three-day starvation causes a decrease in the amount of DCI in muscles by 20%, which may contribute to the insulin resistance after short-term hunger [28]. DCI also affects the regulation of glucose uptake and glycogen synthesis [29].

MI under the influence of epimerase is transformed into DCI. It has been observed that in humans and animals with type 2 diabetes, the amount of DCI excreted in the urine decreases even 10-times, both in patients with non-insulin-dependent and insulin-dependent diabetes, while MI increases [12,13,28,45,46]. Therefore, it is postulated that in diabetes, there may be a secondary defect of epimerase, which impairs the physiological conversion of MI to DCI, stimulated by insulin. The amount of DCI in cells is also reduced by up to 50% [12,13,45,46]. Larner et al. [12] observed this phenomenon for the first time in rats. The physiological conversion is significantly reduced in insulin-sensitive tissues (liver, muscle, fat), and it can be a measure of insulin resistance [4,12,27].

Glucose transporter-4 (GLUT4) is the main of insulin-responsive glucose transporter isoform which plays a key role in the transporting extracellular glucose into insulin-sensitive cells of skeletal muscle and adipose tissues in vivo. In the insulin-sensitive cells, the glucose is decomposed or synthesized to glycogen to maintain normal glucose tolerance [47]. In case of insulin resistance and T2DM the expression of GLUT4 is significantly reduced, impairing the supply and absorption of glucose into the cell [48] (Figure 3).

DP and MI affect the activation of the glucose transporter—GLUT4, which plays a crucial role in the regulation of insulin-stimulated glucose transport to the skeletal muscle and adipose tissue. Binding insulin to its receptor leads to the translocation of GLUT4 from the inside of the cell to its surface with the participation of the phosphatidylinositol 3-kinase (PI3K) signaling pathway [26,47,49]. DP reduces glucose and decreases insulin resistance, but in the presence of PI3K inhibitor, this effect is suppressed [50,35]. In contrast, *myo*-inositolphosphonate (MI-IPG), a derivative of MI, plays a key role in reducing the release of free fatty acids (FFA) from adipose tissue, inhibiting the enzyme adenylate cyclase. In fact, it is known that FFA stimulates insulin resistance and increases triglyceride synthesis. In addition, MI and DCI promote the synthesis of glycogen, inducing the conversion of glucose to glycogen stored inside cells [27,34]. The most recent data indicate that DCI directly specifically stimulates the secretion of insulin in the pancreas [38].

Cyclitols inhibit fat storage in adipose tissue cells, leading to more efficient penetration of glucose into the cells and thus reduction of insulin resistance [35].

Pereira et al. [51] evaluated the effect of ethanolic extract from *Hancornia speciosa* leaf on the hyperglycemic effect and glucose uptake in vitro. This extract consist of, among others, bornesitol and flavonoid glycosides. It caused the inhibition of α-glucosidase activity [51]. The inhibitor of this enzyme is commonly used to treat diabetes as acarbose. It inhibits α-glucosidase, an enzyme found in the initial small intestine, responsible for the breakdown of complex carbohydrates into simple carbohydrates, resulting in a rapid increase in blood glucose. Under the influence of acarbose, the action of this enzyme slows down and its place of action changes—It works on the longer part of the intestine. It decreases the postprandial, rapid increase in blood glucose, while increasing the uptake of glucose in adipocytes [51].

Bates et al. [35] assessed the effect of DP derived from *Bougainvillea spectabilis* on a streptozocin-induced diabetes mice model. The glucose level in plasma and muscle cells in normal mice, diabetes mice and mice with associated obesity after oral and intraperitoneal administration of DP was measured. A decrease in hyperglycaemia above 20% after 6 h was observed. However, the insulin level did not change. Incubation of muscle cells with DP increased glucose uptake by 41% after 10 min and by 34% after 4 h [35]. Similarly, Gao et al. [49] confirmed these data and showed improved glucose tolerance and weight reduction, depending on DP dose [49]. Dang et al. [26] showed that DP improves glucose uptake by activating GLUT4. The administration of DP and MI in mice influenced GLUT4 translocation in skeletal muscle and the reduction in plasma glucose and insulin, which improved insulin sensitivity in skeletal muscle, adipose tissue and liver [26].

Additionally in diabetes, increased endogenous hepatic glucose production is observed, resulting in a further increase in blood glucose. It has been noticed that DP (isolated from soybeans) reduces hyperglycemia in experimental streptozocin-induced diabetes mellitus in rats [17]. The effect of orally administered DP (50 mg/kg body weight over 30 days) was compared with gliclazide, a standard hypoglycemic drug. In DP group, there was a significant reduction in blood glucose, glycosylated hemoglobin (HgbA1), increase in insulin levels, and surprisingly body weight. In addition, the hepatic activity of gluconeogenesis-promoting enzymes (pyruvate kinase, hexokinase, glucose-6-phosphate dehydrogenase) increased, and glycogenolytic enzymes (glucose-6-phosphatase, fructose-1,6-bisphosphatase, lactate dehydrogenase, and glycogen phosphorylase) were reduced [17].

Baquer et al. [39] used fenugreek seed powder orally in rats and confirmed the effect of blood glucose normalization, almost comparable to that of insulin. Fenugreek is a source of DP [52]. After the addition of small traces of metals, additive effects of vanadium were found, while manganese showed additive effects with insulin in vitro in animals with diabetes. The addition of fenugreek eliminates the toxic effects of vanadium, while normalizing the activity of metabolic pathways of enzymes involved in the metabolism of glucose, glucose transport (GLUT4) and insulin level [39]. After the analysis, different effects were found depending on the organs: in the liver the stimulation of glycolysis, lipogenesis, and the decrease in gluconeogenesis, while in the kidneys the processes were reversed. In addition, an improvement in the lipid profile and, what is interesting, an increase in body weight was observed [39]. In fenugreek leaves, stems, pods and seeds d-pinitol is a major cyclitol, representing 43–94% of total cyclitols and 2–77% of total soluble carbohydrates. In dry mature seeds, both the free d-pinitol and its α-d-galactosides (mono-, di- and tri-galactosyl pinitol A) occur at the concentration 0.4–0.6% of dry mass [52].

The hypoglycaemic effect of DCI (obtained from buckwheat, as a natural cyclitol source) in rats was also demonstrated. In a study by Kawa et al. [53], the effect of buckwheat concentrate on hyperglycemia and glucose tolerance in rats with streptozocin-induced diabetes was evaluated. The administration of the concentrate, at 10 and 20 mg of DCI on kg of body weight, decreased serum glucose by 12–19% after 90 and 120 min. These results confirm the usefulness of DCI in diabetes treatment [53]. In addition, the buckwheat concentrate contains MI, also identified as the active ingredient in insulinomimetic [53]. Similarly, Fonteles et al. [54] demonstrated that a single dose of DCI (15 mg/kg) injected into the jugular vein results in a decrease in plasma glucose of diabetic rats by 21%.

There are also a lot of evidences for the beneficial hypoglycemic effect of cyclitols in human. A Korean study by Kang et al. [40] showed the effect of DP from soy on postprandial blood glucose in patients with type 2 diabetes. The study group consumed cooked white rice with DP at a dose of 1.2 g at 0, 60, 120, or 180 min after meal was assessed glycaemia. Postprandial glucose levels were lower after supplementation [40]. Another Korean study—Randomized, double-blind, and placebo-controlled—Confirmed the beneficial effect of soybean DP on the glycemic profile and cardiovascular risk factors in patients with type 2 diabetes [41]. A total of 30 patients received an oral dose of 600 mg DP or placebo twice daily for 13 weeks. In the group subjected to DP supplementation, the mean fasting glycemia, insulin level, fructosamine, HbA1c, and HOMA-IR decreased significantly. Moreover DP caused a significant reduction in total cholesterol, LDL cholesterol, and systolic and diastolic blood pressure, but HDL cholesterol increased. These data suggest that soybean DP may be beneficial in reducing cardiovascular risk, which a consequence of type 2 diabetes [41]. The efficacy and safety of MI and DCI treatment in type 2 diabetes was presented by Pintaudi et al. [55]. This was a three-month oral supplementation of MI 550 mg and DCI 13.8 mg. After 3 months, there was a significant reduction in HbA1c compared to baseline, despite the short treatment period. There was no significant difference in the blood parameters of blood pressure, lipid profile, and BMI. None of the participants reported side effects [55]. In contrast, the Spanish study confirms the hypoglycaemic effect of DP in humans, but at a dose not less than 6.0 g [36].

Lambert et al. [37] evaluated the effect of having a drink containing natural DP from carob pods compared to a sucrose-enriched drink with a glycaemia profile in healthy volunteers, insulin resistance sufferers and type 2 diabetes. Six weeks of taking of DP-enriched beverage resulted in increased production of proteins involved in the insulin secretion pathway only in people with insulin resistance, not in healthy ones [37]. When assessing carbohydrate metabolism in rats, they proposed a new mechanism to improve this metabolism by administering DP. It is a molecule similar to glucose, which is absorbed in the small intestine and distributed through the bloodstream to the entire body. It causes an increase in the amount of complement protein C4A after having a large amount of glucose, correlating with the concentration of C-peptide, which indicates increased secretion of insulin. In addition, C4A protein has a protective effect on β cells of pancreatic islets [37,56].

Insulin-like growth factor-1 (IGF-1) is associated with C-peptide secretion and pancreatic β-cell regeneration, and its low levels are associated with reduced insulin secretion. It can be an important marker of their function [37,57]. Patients with insulin resistance and type 2 diabetes have reduced levels of IGF-1 in serum. IGF-1 is mainly found in the complex with glycoprotein—ALS (acid labile subunit), which is an essential ingredient to maintain its integrity. Human ALS deficiency is characterized by a significant reduction in IGF-1 levels and insensitivity to insulin [37,58]. Regular consumption of a drink with DP induced a significant increase in ALS in people with insulin resistance. There was also a significant increase in the expression of GLUT2 in the jejunum. This glucose transporter is found in the cells of the small intestine, pancreas, liver, and brain. However, no increased expression of GLUT4 and GLUT5 was found [37].

An interesting study was conducted on a group of 110 overweight pregnant women who were randomly assigned to 3 groups using: Revifast^®^ (*Polygonum cuspidatum* extract) with DCI/MI (group I), DCI/MI alone (group II), or placebo (group III) for 30 and 60 days. In the first group, the best results were obtained for the improvement of the glucose profile, total cholesterol, LDL, and TG [59]. This was recently confirmed by Dell’ Edera et al. [60] administering all pregnant women in the first trimester of pregnancy with fasting plasma glucose >92 mg/dL a supplement consisting of DCI 250 mg, MI 1.75 g, zinc 12.5 mg, methylsulfonylmethane 10 mg, and 5-methyltetrahydrofolic acid 400 μg daily, compared to the group received only folic acid 400 μg. More than 3-fold lower exposure of treated women to gestational diabetes, fetal macrosomia was found, but no differences in preterm delivery rate and neonatal hypoglycaemia [60]. This is very important because impaired glucose metabolism during fetal development may contribute to the development of type 2 diabetes in adulthood [6].

Despite the beneficial effects of the cyclitols that increase insulin sensitivity, unfortunately, there are no reports that negate these properties. Davis et al. [61] administered oral soybean DP and DCI (20 mg/kg) to obese subjects (BMI > 30) with type 2 diabetes treated with diet or intolerance of glucose for about a month. Despite the increase in plasma cyclitols level, no increase in insulin sensitivity was observed [61].

Most studies on insulin resistance and associated obesity affect women and adults. In contrast, Mancini et al. [62] evaluated the population of children, boys aged 7 to 15 years, with an average BMI of 29. It was shown that a supplement containing MI 1100 mg, DCI 27.6 mg and folic acid 400 μg was effective in improving insulin sensitivity in children with insulin resistance, although it did not reduce weight [62]. However, data supporting the beneficial influence of DCI in the regulation of body mass, through the central regulation of food intake can be found. Jeon et al. [63] demonstrated the effect of DCI on hypothalamic insulin signaling, by increasing the expression of anorexigenic neuropeptides (proopiomelanocortin) and orexigenic reduction (neuropeptide Y). After injection of DCI derivative, the intake of food and body weight in mice decreased [63].

In conclusion, there is a significant disturbance of the cyclitols in diabetes, which is characterized by:
Abnormal, low DCI levels in urine, plasma, and tissues (liver, muscles, fat),Higher urinary excretion of MIMI deficiency in tissues insensitive to insulin (kidneys, sciatic nerve, lens, retina) [13].

DCI deficiency exacerbates insulin resistance in the liver, muscles, and fat while depletion of MI results in the development of diabetic microvasculopathy (neuropathy, nephropathy and retinopathy) [13].

Summing up, cyclitols by normalizing GLUT activity and improving glucose transport into the cells, protection of pancreatic β-cells and stimulating them to produce insulin can improve carbohydrate metabolism (Figure 4).

## 3. Cyclitols, Polycystic ovary syndrome (PCOS), and Fertility

Polycystic ovary syndrome (PCOS) consists of metabolic disorders and sex hormones, which leads to infertility. Insulin resistance and the resulting hyperinsulinemia contribute to the development of hyperandrogenism, another characteristic for this syndrome. Insulin directly stimulates the ovarian cells to produce more androgens and inhibits the hepatic synthesis of SHBG (sex hormone binding protein), thus indirectly increases the level of circulating free androgens, especially free testosterone [13,44,45,46]. The obvious consequence of insulin resistance is the development of type 2 diabetes, notably with family predisposition [27]. Other features of this syndrome include menstrual cycle disorders, infertility due to hyperandrogenism and obesity. In addition, patients suffer from excessive hair growth on the face and torso, hair loss on the head and acne [27].

There are a lot of articles confirming the beneficial effect of cyclitols in PCOS. Their mechanism of action consists not only in the correction of insulin resistance, but secondarily they also have antiandrogenic effects, which results in the regulation of the menstrual cycle and fertility.

The first study confirming the efficacy of 1200 mg DCI supplementation for 6–8 weeks in PCOS was published by Nestler et al. [64]. Improvements in insulin sensitivity, increasing SHBG, decreasing testosterone levels, plasma triglycerides, normalization of systolic blood pressure, and stimulation of ovulation were observed, compared to the control group [46,64,65].

As in people with type 2 diabetes, women with PCOS have a relationship between insulin sensitivity and DCI-IPG, and improvement after supplementation with DCI and MI. Gerli et al. [66] conducted a randomized, double-blind, placebo-controlled trial in 283 women with PCOS. The incidence of ovulation increased almost twice in women who received DCI with increasing of HDL cholesterol and insulin sensitivity. Similar results were found after oral administration of MI, which is precursor of DCI in vivo [6,59,60]. Cheang et al. [44] described the results of a randomized, controlled trial of DCI supplementation compared to placebo in 11 women with PCOS within 6 weeks. In all subjects, a strict correlation of insulin sensitivity with the release of bioactive DCI-IPG was found. Therefore, DCI-IPG may become the target for therapeutic interventions in PCOS [44]. Interestingly, the use of higher doses of DCI did not confirm these beneficial effects (no improvement in insulin sensitivity) [44]. These researchers have risked the idea that insulin resistance in women with PCOS develops as a result of DCI-IPG deficiency. The obese women with and without PCOS were compared, noting that the bioactivity of DCI-IPG was reduced only in the group of patients, regardless of obesity and weight loss [44]. Similar results were observed in the group of obese patients treated with MI and better therapeutic effects were obtained at the beginning. However, there was no reduction in body weight [11,67].

In Polish studies, patients with PCOS received MI at a dose of 4 g/d [8]. In the whole group a statistically significant decrease in LH/FSH and HOMA-IR coefficients was observed, both in the first and third month of the clinical trial. In 81% of patients ovulation and spontaneous menstruation occurred within 3 months of treatment [11].

It has been proven that the effect of DCI in PCOS is to reduce the expression of CYP19A1, P450scc and the receptor for IGF-1, in a dose-dependent manner. This affects the inhibition of enzymes involved in steroidogenesis in ovaries, stimulated by gonadotropin hGCS, which is stimulated by insulin. Insulin is responsible for increasing the activity of CYP19A1 and P450scc [46].

Piomboni et al. [68] also found a significant reduction in oxidative stress in patients with PCOS who underwent ovarian hyperstimulation with supplementation prior to treatment with DCI and/or metformin. These drugs have shown significantly improvement vesicular fluid environment, reducing oxidative damage, which consequently increases the number of valuable oocytes [68]. Good quality oocytes require proper balance in pro-oxidative and antioxidant factors, because oxidative stress is associated with low levels of fertilization and atresia of follicles. Hence, they generate new research ideas on the use of cyclitols in women with endometriosis who also suffer from infertility and oxidative status disorders [68]. However, Dinicola et al. [69] point out that DCI alone, when administered at high doses, adversely affects the quality of oocytes and ovarian response, and the combination therapy of MI with DCI gives better clinical results [69]. It has been hypothesized that the activity of epimerase, which converts MI into DCI, increases in the ovaries of patients suffering from PCOS and that the resulting deficiency of MI is responsible for low quality of oocytes [6,70].

The available literature on this topic analyzed the impact of MI on the controlled hyperstimulation with the use of gonadotrophins in women with PCOS. Women receiving MI required a significantly lower total dose of gonadotropins for stimulation, which reduced the risk of ovarian hyperstimulation syndrome. In those patients who were prepared for extracorporeal fertilization programs, who received MI and DCI, had higher quality of oocytes and embryos, as well as a greater number of reported pregnancies after the transfer [11,71].

In PCOS, the use of DCI improves insulin sensitivity, while MI affects FSH production and oocyte quality [45,46]. This was once again confirmed by Ciotta et al. [72] who assessed the influence of MI on the quality of oocytes in women with the syndrome. Administration of 2 g of MI and 200 micrograms of folic acid for 3 months significantly increased the number of vesicles germ and oocytes in the ovaries, improved the quality of the embryos [72]. Benelli et al. [73], in addition to improved metabolic rates, found a significant reduction in LH, free testosterone, and a significant increase in 17-beta-estradiol in the group of women treated with combined MI with DCI (550 mg of MI, 13.8 mg of DCI, 200 μg folic acid) [73]. Another study highlights the importance of cyclitols for the conditions of fertilization and the development of a fertilized zygote. A positive correlation was found between MI and estradiol concentration with a better oocyte developmental potential [74]. The ratio of MI to DCI was much higher in good quality blastocysts in mice and the content of DCI in vesicles germ fluid was lower. MI supplementation is a hope for women undergoing in vitro fertilization because it may improve the quality of oocytes [74].

Unfer et al. [34] presented a meta-analysis regarding the effect of MI alone or a combination with DCI on hormone profile and metabolic disorders in women with PCOS. Only randomized trials were considered, including nine papers (247 patients and 249 controls). MI was shown to be effective in reducing fasting insulin and HOMA, reducing testosterone levels with unchanged androstenedione. However, a significant increase in serum SHBG was observed only in those studies in which MI was administered for at least 24 weeks [34].

To sum up, treatment with cyclitols of PCOS is also effective in reducing BMI, even without lifestyle modification [29,67]. DCI affects the normalization of LH, LH/FSH ratio. Administration of MI is more effective in obese patients who have high fasting serum insulin or hyperinsulinemia and familiar predisposition to diabetes [29,67]. No side effects of the therapy are observed, and general results provide evidence of IA level [64,75]. The value and effectiveness of supplementation with cyclitols has been appreciated and recommended by the Polish Gynecological Society in 2014 [11].

Recently, research has been conducted on the potential use of MI in the treatment of male infertility. Antioxidative, prokinetic, and hormonal regulation may be a chance for men with astezozospermia (abnormal motility of spermatozoa in the ejaculate) [76]. Physiologically, the concentration of MI produced by Sertoli cells in response to FSH is significantly higher in seminal tubes than in serum. It participates in processes involving the regulation of motility, capacity, and response of sperm akrosome, probably regulating the osmotic concentration [76].

## 4. Cyclitols and Hypertension

Hypertension leads to damage of myocardium, strokes and kidney failure. The pathophysiology of hypertension includes the complex interaction of many vascular effectors, including the activation of the sympathetic nervous system, the renin-angiotensin-aldosterone system and inflammatory mediators. There is a narrowing of the vessels and inflammation, which leads to the rebuilding of the vessel wall and eventually to the formation of atherosclerotic lesions, the consequences of advanced disease. As a result of inflammation, oxidative stress reduces the bioavailability of natural vasodilatant nitric oxide and causes endothelial dysfunction [77]. Drugs with these properties are widely used in the treatment of hypertension.

Silva et al. [78,79] investigated the effect of the *Hancornia speciosa* Gomes leaf plant (*Apocynaceae*), commonly found in Brazil, on the cardiovascular system in rats. It has been observed that the ethanolic leaf extract of this species affects vasodilation on the mesenteric muscle of vessels due to the production of nitric oxide in the endothelium by activation of PI3K. The activity of ACE (angiotensin converting enzyme) in serum and angiotensin II levels were significantly reduced by plant extract, compared to the control group treated with traditional antihypertensive drug (captopril). This extract contains flavonoids and bornesitol [78,79,80]. Another study by the same authors confirmed previous observations. Surgical removal of the kidney and giving deoxycorticosterone induced hypertension in the examined mice. In mesenteric vessels, vasodilation was measured with a myograph. The extract lowered the blood pressure, caused arterial vasodilatation, depending on the dose with the increase in plasma nitrite [78].

## 5. Cyclitols and Atherosclerosis

Cyclitols have an anti-atherosclerotic effect. OxLDL (oxidized low-cholesterol molecules) are absorbed by macrophages which form in foam cells. They build an atherosclerotic plaque that closes the vessels causing myocardial infarction. DP inhibits the foam cells formation, thereby limiting atherosclerotic plaque (Figure 5). OxLDL is also an antigen in atherogenesis that stimulates the immune mechanism of inflammation [81,82]. The early infiltration of the internal vascular membrane by macrophages and their activation is the first stage in the development of atherosclerotic lesions. Due to scavenger receptors present on their surface they store ox-LDL, they form foam cells, pathognomic for atherosclerosis in the form of fatty streaks. In addition, ox-LDLs become antigens that promote the differentiation of macrophages into dendritic cells that present antigen and activate T-cells. In the next stage, macrophages release enzymes, mainly matrix metalloproteinases, which break down the connective tissue. The next step is to create fibrous lesions as a result of connective tissue splitting, characterized by the accumulation of lipid-rich remnants of smooth muscle cells [83,84].

Choi et al. [82] studied the effect of soybean DP on the formation of foam cells using differentiated THP-1 macrophages (a human monocytic cell line derived from an acute monocytic leukemia). It was confirmed that DP dose-dependent inhibits the formation of ox-LDL. Lower release of TNF and chemotactic protein for monocytes-1 (MCP-1) after DP administration (0.05–0.5 mM) was observed. IL-1β secretion, IL-8 and gene 9-M-metaloproteinase expression in the matrix were significantly reduced even at low doses of DP (0.05 and 0.1 mM) compared to the cells to which it was not added. It confirms its strong anti-atherosclerotic effect [82].

In preventing the development of atherosclerosis, an important element is the improvement of the lipid profile. After oral administration of DP and MI, the lipid profile is improved with a decrease in serum triglyceride and total cholesterol [41,81,82].

## 6. Cyclitols for Nervous System Diseases

Inositol and its derivatives have signaling and regulatory functions in many cellular processes, including the nervous system. In mammals, the concentrations of inositol in plasma is 0.03 mM, while intracellular are several times higher than in the circulation, in cerebrospinal fluid 0.2 mM and the highest occurs in the brain—6 mM [85,86].

Disorders of MI metabolism to phosphates and its secondary accumulation occur, among others, in bipolar disorders, Alzheimer’s disease, and epilepsy [86]. However, the decrease MI concentration was observed in patients undergoing drug therapy, along with an improvement in the clinical condition [86]. The therapy with lithium and valproic acid caused changes in the concentration of inositol in the frontal and temporal lobes, cingulate of the rim and basal ganglia, resulting in the reduction of its biosynthesis [86,87].

Inositol imbalance is also associated with a number of neurodegenerative diseases, including Alzheimer’s disease. In patient’s brain, there is an increased level of MI compared to healthy individuals, despite different pathogenesis of the disease, connected with the accumulation of neurotoxin-toxic amyloid A [86,87]. Liu et al. [88] evaluated the brains of mice after 6 months of *scyllo*-inositol supplementation, observing the lower amyloid A content, the greater activity of microglia and the phagocytosis capacity compared to non-treated mice. *Scyllo*-inositol binds and stabilizes small soluble conformations of Aβ amyloid that are phagocytosed by microglia [88].

Elevated concentration of inositol also occur in regions of the brain that are involved in seizures, up to 20% in the hippocampus, suggesting an attack-induced alteration of inositol signaling in the brain [86,30]. Pascente et al. [30] observed a transient increase in the concentration of MI in the hippocampus in pilocarpine-induced epilepsy with accompanying loss of astrocytes [30].

In turn, Greek researchers described newly identified autoantibodies associated with cerebellar disorders. Sera of 15 patients with cerebellar ataxia on the mouse brain substrate were examined. Antibody against receptor 1 for inositol in Purkinje neurons has been identified. Since mice with a congenital defect have symptoms of ataxia and epilepsy, the autoantibody may have a functional role. This gives the opportunity to develop specific immunotherapy in the future in these disorders [31].

Inositol deficiencies may lead to development of abnormalities. Based on the research, inositol with folic acid showed a greater protective effect on the development of the nervous system in the fetus than supplementation with folic acid alone [31].

## 7. Cyclitols and Carcinogenesis

Normal cells in the body undergo a natural, programmed death (apoptosis). Cancer cells do not undergo apoptosis, becoming “immortal”, which contributes to rapid progression. The anti-cancer effect of DP has been proven [24]. It stimulates the mechanisms of apoptosis in cancer cells.

Al-Daghri et al. [89] evaluated the effect of fenugreek extract on Jurkat cell. Jurkat cells are a model of immortalized human T lymphocytes to assess the susceptibility of tumors to drugs and radiation. Cytotoxic effect of tumor cells was observed in the form of stimulating autophagy, depending on the dose and duration of action [89]. The next study also used the fenugreek extract, investigating its effect in vitro on the cancer cell lines, including T-cell lymphoma and normal cells, using different concentrations (100 μg/mL, 200 μg/mL, 300 μg/mL) in specific time (24 h, 48 h, 72 h and 96 h). Selected cytotoxic induction of apoptosis was observed, involving only abnormal cells. Normal lymphocytes did not show apoptosis [90]. The authors were inspired by a case report of a child with brain T-lymphoma and secondary hydrocephalus, which after the initial remission due to chemo- and radiotherapy had a recurrence of the disease. The family gave a concentrated fenugreek extract (about 8 g fenugreek seeds per day) without medical consultation for 6 months. MRI of the brain after six months showed complete remission which observed for 11 years [91]. Apoptosis induced by fenugreek was described in many different tumor cell lines: colon, osteosarcoma, leukemia, breast, and liver [90,92]. Shabbeer et al. have shown that fenugreek is cytotoxic to cancer cells, but not to healthy cells. Administration of 10–15 μg/mL fenugreek extract for 72 h inhibited the growth of breast, pancreatic, and prostate cancer cell lines [93].

Rengarajan et al. [24] confirmed the antiapoptotic effect of DP, derived from soy, which was concentration-dependent and it induced p53 and Bax expression, and inhibited Bcl-2, Bcl-XL in the MCF-7 cell line (a breast cancer model). Reduction of glutathione, increase in LDH in MCF-7 was observed, confirming its cytotoxic effects (40 and 60 μM concentrations) [24]. Inhibition of glutathione production in cancer cells reduced their resistance to cytostatic drugs, because this enzyme protects against oxidative stress. Its deficiency causes damage to cells in the mechanism of oxidative stress-induced cytotoxicity. Deficiency of endogenous glutathione may stimulate apoptosis by inducing proapoptotic factors [24]. Similar results were obtained by Lin et al. [23] in prostate cancer, observing the suppression of metastases. Administration of DP caused the reduction of mRNA and the expression of integrin αvβ3 on the cell surface. Integrins are the major adhesive molecules involved in the formation of metastases. In addition, DP exerted an inhibitory effect, reducing the phosphorylation of focal adhesion kinase (FAK), c-Src kinase activity and NF-kB activation, pathways responsible for, among others cell mobility [23].

## 8. Cyclitols and Allergic Diseases

The anti-inflammatory properties of cyclitols and the ability of DP to enhance the expression of the transcription protein GATA-3, the key molecule regulating Th1/Th2 balance, can be used in the treatment of allergic diseases (atopic dermatitis, asthma) [94]. Bae et al. [94] demonstrated the effectiveness of fenugreek extract in mice in induced contact dermatitis and hypersensitivity to ovalbumin (250 mg/kg body weight of fenugreek extract for 7 days after sensitization). It reduced of inflammation, decreased the amount of eosinophils and mast cells in the infiltration. Moreover it inhibited the production of IL-4, IL-5, IL-13, and IL-1β. Fenugreek extract prevented the differentiation of Th2 cells in splenocytes of mice with allergy, reducing the secretion of IL-4 and expression of GATA-3 mRNA, an IL-4 transcription factor. In result they observed normalization of IFN-γ secretion and number of IFN-γ producing Th1 cells [94].

In contrast, the extract of *Argyrolobium roseum*, a Himalayan plant containing DP, in a dose-dependent manner inhibited antibody formation and delayed type hypersensitivity reactions, reducing the number of CD3, CD19, CD4, and CD8 lymphocytes and expression of Th1/Th2 cytokines in splenocytes [95]. At the same time, they showed a large safety profile and no toxicity, despite immunosuppressive properties, both in vitro and in vivo in Swiss mice and Wistar rats, which received 300–1200 mg/kg of body weight pinitol orally [95].

## 9. Adverse Effect of Cyclitols

In adults, inositol in doses of up to 6 g to as much as 18 g did not cause major side effects, except mild bloating or diarrhea in a few patients. No other side effects were reported [13,33]. Supplementation with inositol (2 × 2 g per day) throughout the entire pregnancy from the first trimester was not associated with side effects or increased risk of adverse effects of pregnancy [13,33]. This was confirmed in the studies on pregnant women and on males, and also on newborns [59,60].

## 10. Conclusions

The anti-inflammatory properties of cyclitols may be a good therapeutic option in the reduction of metabolically induced inflammation—Metainflammation, which leads to more frequent development of metabolic disorders (diabetes, hypertension) and atherosclerosis [9]. *Myo*-inositol and other cyclitols have several health-promoting and therapeutic properties such as: improving lipid profile in decreasing of serum triglicerydes and total cholesterol, as well as having an insulin-mimetic effect. Moreover, they have antioxidative, anti-inflammatory, and anti-cancer properties [9]. Due to well drugs tolerance and low toxicity of these compounds, cyclitols are recommended for pregnant women and also for children.

The widespread presence of cyclitols in nature and their availability predisposes them to be used beyond medical applications. Plant-derived food containing these compounds may in the future play an important role in the prophylaxis of the development of lifestyle diseases such as T2DM, neoplasmas and allergies diseases.

## Figures and Tables

**Figure 1 nutrients-10-01891-f001:**
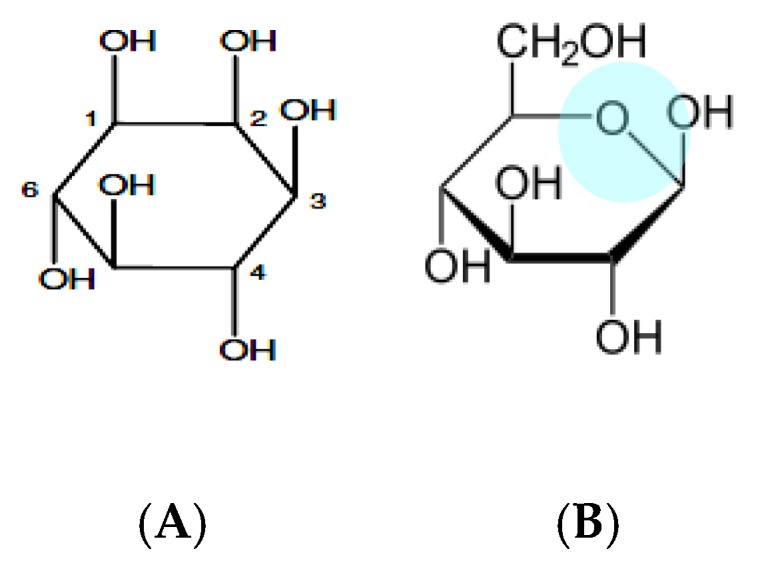
Example of a molecular structure of cyclitol (**A**) and sugar (**B**) [1].

**Figure 2 nutrients-10-01891-f002:**
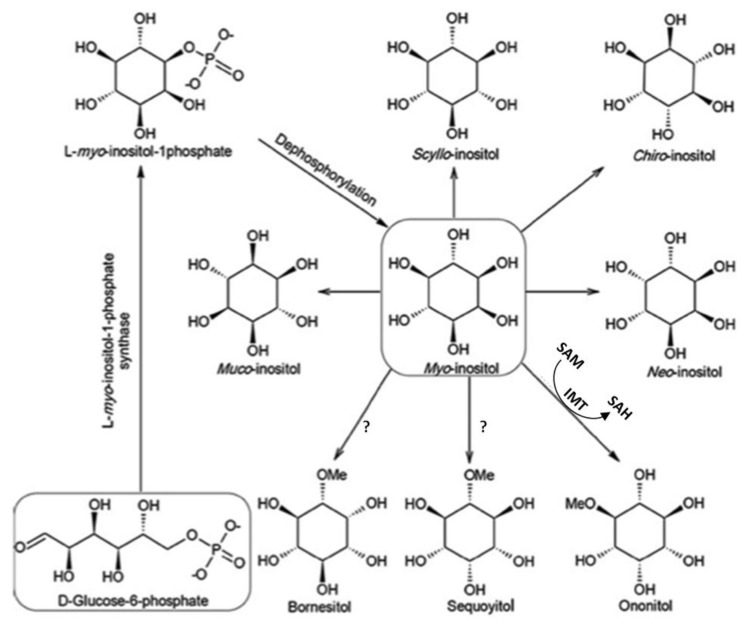
The biosynthetic pathway of *myo*-inositol and its methylated derivatives. The biosynthetic pathway of *myo*-inositol and its methylated derivatives. IMT—*myo*-inositol *O*-methyltransferase, SAM—S-adenosylmethionine, SAH—S-adenosylhomocysteine [10].

**Figure 3 nutrients-10-01891-f003:**
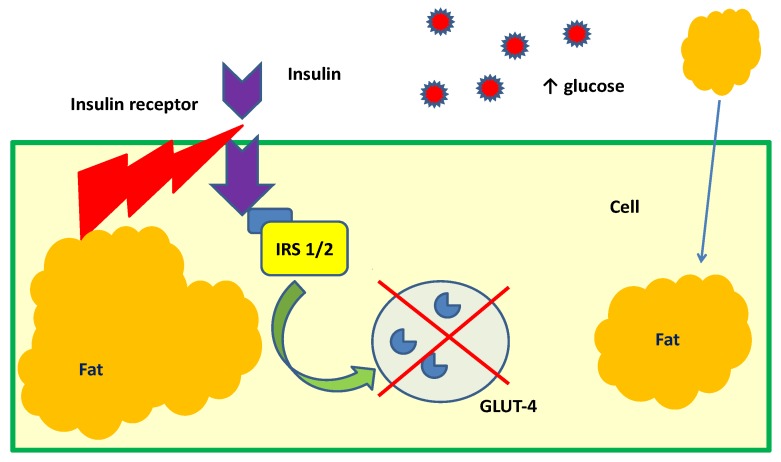
Mechanism of insulin resistance in the cell [27].

**Figure 4 nutrients-10-01891-f004:**
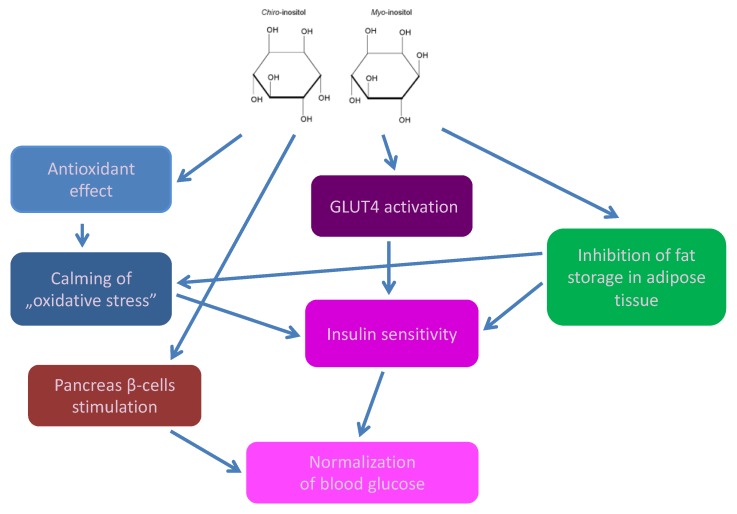
The action of cyclitols in diabetes mellitus.

**Figure 5 nutrients-10-01891-f005:**
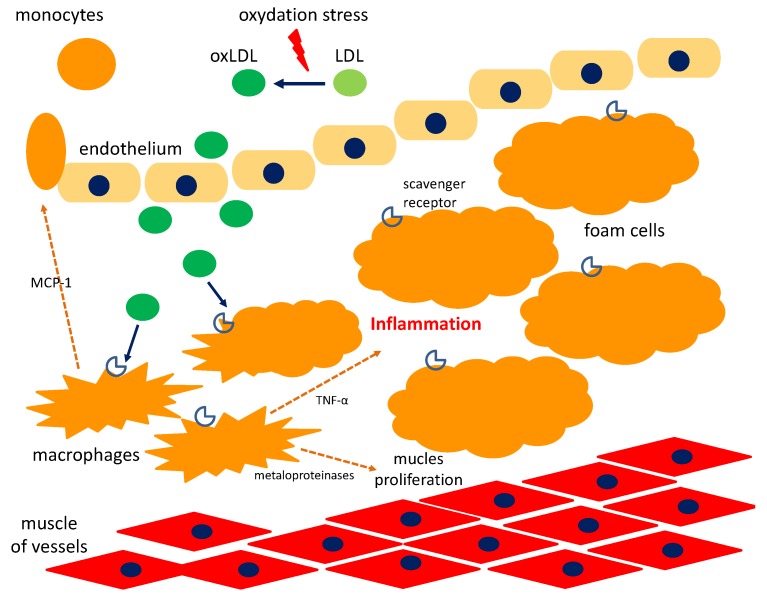
Atherosclerotic plaque formation.

**Table 1 nutrients-10-01891-t001:** The effect of cyclitols on animals’ metabolism.

NO.	Authors	Cyclitol/Source	Group	Effect
1	Pereira et al., 2014 [26]	Bornesitol/ (*Hancornia speciosa*)	epididymal adipocytes of Swiss mice	—Inhibition of alpha-glucosidase activity —Increase in glucose uptake in adipocytes
2	Sivakumar & Subramania, 2009 [12]	d-pinitol (chemical pure)	Rats with STZ, d-pinitol 50 mg/kg body weight, 30 days	—Decreasing of blood glucose, HgbA1, increasing of insulin levels and BMI —Increasing of hepatic gluconeogenesis enzymes activity and decreasing of glycogenolytic enzymes activity
3	Bates et al., 2000 [27]	d-pinitol/ (*Bougainvillea spectabilis*)	Obese mouse (ob/ob) with STZ, intraperitoneal and oral d-pinitol 100 mg/kg body weight	—Decreasing of blood glucose, without changes in insulin level —Increasing of glucose uptake by muscle cells
4	Dang et al., 2010 [28]	d-pinitol and *myo*-inositol	DP or MI 1 g/kg body weight orally in mice for 30 min before oral administration of glucose (2 g/kg of body weight)	—Increasing of GLUT4 translocation in skeletal muscle —Reduction of plasma glucose and insulin levels
5	Gao et al., 2015 [29]	d-pinitol	Insulin resistance induced by diet and STZ in Sprague Dawley rats, d-pinitol in two doses (30 and 60 mg/kg body weight)	—Decreasing of fasting blood glucose by 12.63% in the high-dose group —Improvement of oral glucose tolerance
6	Geethan et al., 2008 [30]	d-pinitol	Sprague Dawley rats (STZ)—DP 25, 50 and 100 mg/kg body weight	—Decreasing of Low-density lipoprotein (LDL) and Very-low-density lipoprotein (VLDL), a significant High-density lipoproteins (HDL) increasing in serum
7	Hannan et al., 2006 [31]	fraction of *T. foenum-graecum*	STZ rats, a soluble dietary fibre fraction of *T. foenum-graecum* (0.5 g/kg body weight)	—Decreasing of glucose in serum, increasing of glycogen content in the liver —Increasing of total antioxidant status —No effect on insulin secretion
8	Kawa et al., 2003 [32]	d-*chiro*-inositol/ chemically synthesized	STZ rats fed with buckwheat concentrate (10 and DCI 20 mg/kg body weight)	—Reduction of glucose by 12–19% after 90 and 120 min in the serum after administration of the concentrate

**Table 2 nutrients-10-01891-t002:** The effect of cyclitols on humans’ metabolism.

No.	Authors	Cyclitol/Source	Group	Effect
1	Davis et al., 2000 [33]	soybean-derived DP—20 mg/kg/day	22 obese subjects (BMI 36.6) with diet-treated T2DM or glucose intolerance (HbA1 6.8%): 12 receive either DP or 10 placebo in a 28-day double-blinded trial	—No toxicity of DP was observed —Plasma levels of DP were 48-fold and DCI levels 14-fold greater in DP group compared with placebo
2	Kim et al., 2004 [34]	0.6 g soybean-derived DP	30 patients with T2DM received an oral dose of DP or placebo twice daily for 13 weeks	—DP significantly decreased fasting plasma glucose, insulin, fructosamine, HbA1c and HOMA-IR —DP significantly decreased total cholesterol, LDL, the LDL/HDL ratio, and systolic and diastolic blood pressure and increased HDL
3	Kang et al., 2006 [35]	0.6 g DP from soybean	15 subjects with tT2DM ingested cooked white rice containing 50 g of available carbohydrate with or without prior ingestion of DP (1 g dose at 0, 60, 120, or 180 min or as a 0.6 g dose at 60 min prior to rice ingestion)	—1.2 g of DP 60 min prior to rice consumption controlled blood glucose most effectively, significantly diminishing the postprandial increase in plasma glucose levels at 90 and 120 min after rice consumption —DP had no apparent effect on postprandial insulin levels
4	Hernández-Mijares et al., 2013 [36]	a nutritive beverage (Fruit Up^®^) containing 2.5, 4.0, or 6.0 g of DP	31 healthy volunteers, BMI 20–30, fasting glycaemia <100 mg/dL; serum glucose and insulin levels were determined at 0, 15, 30, 45, 60, 90, and 120 min for each dose of DP	—Good tolerated of DP and no adverse events (including hypoglycaemic episodes) for any doses administered —Only a dose 6 g of DP reduced glucose and insulin levels at 45 and 60 min
5	Mancini et al., 2016 [37]	MI 1100 mg, DCI 27.6 m, folic acid 400 µg (Inofolic Combi, Lo.Li. Pharma, Rome, Italy)	23 obese children, aged 7–15, with a mean BMI 29.8, 11 treated with normocaloric diet, physical activity and Inofolic and 12 were controls	—MI/DCI lowered more effective insulin increase after OGGT in children with higher basal insulin level —BMI after six months of supplementation was not significantly different from the baseline which emphasize only inositol role in glucose profile improvement
6	Pintaudi et al., 2016 [38]	MI 550 mg, DCI 13.8 mg, folic acid 400 µg (Inofolic Combi, Lo.Li. Pharma, Rome, Italy)	20 subjects a combination of MI + DCI was suggested to be taken orally twice a day as add-on supplement to hypoglycemic treatment	—After 3 months of treatment fasting blood glucose and HbA1c levels significantly decreased compared to baseline —There was no significant difference in blood pressure, lipid profile, and BMI levels —None of the participants reported side effects
7	Lambert et al., 2017 [39]	a sweetened beverage with natural carbohydrates containing DP (PEB) compared to a sucrose-enriched beverage (SEB)	40 healthy volunteers and 40 overweight volunteers with impaired glucose tolerance (IGT) and 38 T2DM patients, 6 weeks intake PEB or SEB	—A significant increase in two proteins involved in the insulin secretion pathway: IGF acid labile subunit and complement C4A in IGT subjects but not in healthy volunteers —PEB induces changes in the insulin secretion pathway that could help to reduce blood glucose levels by protecting β-cells and by stimulating the insulin secretion pathway
8	Malvasi et al., 2017 [40]	MI/DCI + Trans-resveratrol (Revifast^®^)	104 pregnant (gestational age between the 24th and 28th week) were treated: 35 group I Revifast^®^ + DCI/MI, 34 group II DCI/MI alone and placebo group-35	—After 30 and 60 days of therapy no difference in systolic and diastolic parameters among 3 groups during study —All blood chemistry parameters improved compared to placebo at 30 days already, but significantly to 60 days —Group I demonstrates significantly improved lipid and glucose parameters than group II, which are at 30 to 60 days of treatment —None of the women reported an adverse reaction to therapy
9	Dell’ Edera et al., 2017 [41]	DCI 250 mg/d, MI 1.75 g/d, 12.5 mg/d zinc, 10 mg/d methylsulfonylmethane, 400 μg/d 5-methyltetrahydrofolic acid	40 pregnant with the onset of GDM were treated DCI/MI, from the first trimester of pregnancy, with and 43 controls with only 400 μg/d folic acid	—At the 24th week of pregnancy, the incidence of maternal GDM was lower in the treated group (RR 3.35) —A significant difference was observed between treated and control groups in terms of risk of macrosomia —No significant difference was identified between two groups, regarding neonatal hypoglycemia and preterm delivery

DP—d-pinitol, DCI—d-*chiro*-inositol, MI—*myo*-inositol, GDM—gestational diabetes mellitus; OGGT—oral glucose tolerance test; STZ—streptozotocin-induced diabetes, T2DM—Type 2 diabetes mellitus.

**Table 3 nutrients-10-01891-t003:** Cyclitols—Indications in medicine.

Cyclitol	Application in Medicine
d-pinitol	—Reduction of hyperglycaemia and plasma glucose level —Improvement of lipid profile —Inhibition the formation of foam cells and oxLDL —Inhibition of inflammation and carcinogen-induced activation of NF-κB, which leads to reduction of proliferation, invasion, and angiogenesis in tumors —Stimulation of apoptosis and autophagy of lymphoma cells —Reduction of osteoclastogenesis by inhibiting NF-κB ligand receptor activator (RANKL) —In rats, protection against chemically induced liver damage
*myo*-inositol	—Treatment of type 2 diabetes and insulin resistance (induce translocation GLUT4 into the cell membrane, thereby increasing cellular uptake of glucose) —Stimulation of menstruation and ovulation in polycystic ovary syndrome (PCOS), improvement of oocyte and blastocyst quality —Improvement of osteogenesis —Disturbance of transformation in neurodegenerative diseases (increase in brain concentration)
d-*chiro*-inositol	—Treatment of type 2 diabetes and insulin resistance (participates in insulin signaling, stimulating enzymes involved in the regulation of glucose metabolism) —Improvement of lipid and carbohydrate profile in pregnant women, prophylaxis of gestational diabetes —Appetite regulation—Influence on hypothalamic insulin signaling —Treatment of insulin resistance, hyperandrogenism and infertility in PCOS —Inhibition of osteoclasts, improvement of osteogenesis
*scyllo*-inositol	—Reduction of the content of amyloid A (helps prevent the formation of insoluble amyloid fibrils) and increase the activity of microglia and the ability of phagocytosis in the brain —In rats, reduction of epileptic seizures induced by pentylenetetrazole
bornesitol	—Lowering of blood pressure, dilation of arteries vessels depending on the dose

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
