# Peer review of "The Healing-Promoting Properties of Selected Cyclitols—A Review"

_nutrients, 2018, doi:10.3390/nu10121891_

Reviewer 1 Report

General Comments:

The purpose of this manuscript was to present the health promoting properties of cyclitols. Cell signaling via inositol phosphates, in particular via the second messenger myo-inositol 1,4,5-trisphosphate, and phosphoinositides comprises a huge and  extremely important field of biology. While the beneficial effects of myo-inositol and inositol hexaphosphate (IP6) for human health are recognized, “other” inositols, their phosphates and derivatives are rarer in nature and their potential biological functions are not well-known. Therefore, exploring the role and potential medical applications of these “other” isomers is timely and relevant. The manuscript is original and well organized, with a logical approach.

Specific Comments:

1. Consider changing the title. The “healing” is a strong word and medical term. On the other hand, it sounds like the “voodoo or witch science”. Consider changing “the healing properties” to “the health-promoting properties”.   

2. Carefully check the manuscript for the consistency, typographical errors, and better flow. For example, Ln 19: Reward the sentence “The most known is using of them in ….”  (bad English). Also,

Ln 463: Reward the sentence “Thanks good tolerability and…. “ (bad English).

3. State how many “isomers” or derivatives exist in this huge family of inositols (approximately).

4. Add a list of Abbreviations.

5. Although, this is a review article, not too many review articles are referenced.

6. All figures need a Figure legend and need to be stated/listed/discussed in the text.

7. Redo/recreate Figures 5 and 6, and enrich them with some molecular pathways. Figure 6 (carcinogenesis and apoptosis) is in particular poor.

8. Tables need some clarity and better organization.

9. Rewrite the “Conclusion”. The anti-inflammatory properties of cyclitols is not the only mechanism of action.

10. Add a “future directions”. 

Author Response

The answer for reviewers

Thank you for the time and work devoted to improving the manuscript and valuable suggestions.

Agnieszka Owczarczyk-Saczonek

Reviewer 1:

Specific Comments:

1. Consider changing the title. The “healing” is a strong word and medical term. On the other hand, it sounds like the “voodoo or witch science”. Consider changing “the healing properties” to “the health-promoting properties”.  – I’ve changed: “The healing-promotingproperties of selected cyclitols – a review.”

2. Carefully check the manuscript for the consistency, typographical errors, and better flow. For example,

Ln 19: Reward the sentence “The most known is using of them in ….”  (bad English). Cyclitols are well known compounds in the treatment of an  accompanied diabetes insulin resistance, also obesity and polycystic ovarian syndrome.

Ln 463: Reward the sentence “Thanks good tolerability and…. “ (bad English). Due to well drugs tolerance and low toxicity of these compounds, cyclitols are recommend for pregnant women and also for children.

I’ve corrected the manuscript with my English teacher.

3. State how many “isomers” or derivatives exist in this huge family of inositols (approximately).

Among cyclitols, well known myo-inositol is widely present in  nature. There are nine possible isomers of myo-inositol. Five of them  naturally occur like myo-, scyllo-, muco-, neo- and D-chiro-inositol,  while other four isomers (L-chiro-, allo-, epi-, and cis-inositol) are  derived from myo-inositol and obtained by chemical synthesis [1,2].

1.    Campbell, J.A.; Goheen, S.C.; Donald, P.; 2011. Recent trends for enhancing the diversity and quality of soybean products, in:  Prof. Dora Krezhova (Ed.), Extraction and analysis of inositols and other carbohydrates from soybean plant tissues, E-Publishing, pp. 281-304.

2.    Al-Suod, H.; Ligor, M.; Ratiu, I.-A.; Rafińska, K.; Górecki, R.;  Buszewski, B. A window on cyclitols: Characterization and analytics of inositols. Phytochem Let. 2017,20, 507-519,  doi:10.1016/j.phytol.2016.12.009.

4. Add a list of Abbreviations. – I’ve done

5. Although, this is a review article, not too many review articles are referenced. I’ve added some references.

6. All figures need a Figure legend and need to be stated/listed/discussed in the text. – I’ve done

7. Redo/recreate Figures 5 and 6, and enrich them with some molecular pathways. Figure 6 (carcinogenesis and apoptosis) is in particular poor.I’ve removed Fig. 6 and recreated Fig.5

8. Tables need some clarity and better organization. – I’ve changed them a little bit.

9. Rewrite the “Conclusion”. The anti-inflammatory properties of cyclitols is not the only mechanism of action.I’ve done.

10. Add a “future directions”.  I’ve done.

Reviewer 2 Report

This review discusses what could potentially be an important and interesting topic. Too often reviews on cyclitols discuss only myo-inositol without even mentioning the existence of other cyclitols. Unfortunately, this review does not do justice to the topic. The poor quality of English caused difficulty in comprehending several passages. Language quality is not reason to reject the essay, however, the unfortunate use of English coupled with numerous counts of misinformation and some jaw-dropping statements generate a manuscript that cannot be published by any scientific journal. The authors could improve the manuscript by editing but I recommend writing a completely new paper.

The cyclitols would benefit from a more comprehensive introduction. A chapter should be included on the discovery of known cyclitols and of the enzymology producing them, as well as a chapter dedicated to the myo-inositol phosphates, the most studied cyclitols. In the current manuscript the link between cyclitols and human diseases (starting from chapter 2: diabetes; PICOS; hypertension; atherosclerosis; Alzheimer’s; carcinogenesis; allergy chapter 8) is just a list of useless information, often poorly referenced and/or incorrect. If the authors aim to discuss cyclitols in the context of human disease they should do so properly, starting with explaining the basic biology of these molecules, then integrating these facts with what is known of their tangible use in medicine. The authors review the beneficial use of fenugreek (Chapter 7: carcinogenesis) that may or may not contain D-pinitol. It certainly contains many other molecules, and linking the beneficial use of this plant just to D-pinitol is not science, it is pure speculation.

The slim figure legends are another reason for concern; explaining the meaning of the figures would benefit the reader. For example, Figure 1 should state that cyclitols are not reducing sugars and explain the reason. Figure 2 shows that of the three methyl-derivates of myo-inositol, only one is synthesized from S-Adenosylmethionine with “IMT/catalysis”. From where the other two are synthesized is unclear. It is also not clear what IMT stands for: it is misspelled Massachusetts Institute of Technology or Inositol Methyl Transferase? Nowhere in the paper is IMT defined.

The several tables are not referenced in the main text!  How this is possible?

The two biggest jaw-dropping statements:

Line 63: “inositol triphosphate, a hormone regulating other hormones”

Inositol triphosphate is a cellular second messenger generated after the activation of phospholipase-C. This is a textbook example of signal transduction. Please read Chapter 16 - Essential Cell Biology. 4th Edition. by B. Alberts, D. Bray, K. Hopkin, A. Johnson, J. Lewis, M. Raff, K. Rober. Publisher: Garland Science. ISBN: 9780815344544.

Line 374 “In mammals, the level of inositol in plasma is from 25 to 100 M”

Myo-inositol’s solubility limit is ~1-2 M in water. How is it possible that in the much denser plasma its solubility increased by a factor of 100?

Misinformation:

Line 153 “Baquer et al. [37] used fenugreek seed powder orally (DP [D-pinitol] source) in rats”.

Reference [37] does not state that fenugreek is a source of D-pinitol. Fenugreek as source of D-pinitol appears several times in the text, but there are no references indicating this plant as a source of D-pinitol. Bizarrely, the author’s own work, recently published in Acta Physiologiae Plantarum, is not cited.

In Table 2 is cited Davis et al., 2000 [94].

However, the manuscript list only 93 references.

I could continue listing the many inappropriate statements but I prefer to stop here. I will stress that the above list of imperfections is certainly not exhaustive.

Author Response

The answer for reviewers

Thank you for the time and work devoted to improving the manuscript and valuable suggestions.

Agnieszka Owczarczyk-Saczonek

Reviewer 2:

This review discusses what could potentially be an important and interesting topic. Too often reviews on cyclitols discuss only myo-inositol without even mentioning the existence of other cyclitols. Unfortunately, this review does not do justice to the topic. The poor quality of English caused difficulty in comprehending several passages. Language quality is not reason to reject the essay, however, the unfortunate use of English coupled with numerous counts of misinformation and some jaw-dropping statements generate a manuscript that cannot be published by any scientific journal. The authors could improve the manuscript by editing but I recommend writing a completely new paper. - In our manuscript, we tried to concentrate on the use of cyclitols in medicine, limiting and simplifying data on the basic biology of these molecules because the article would exceed the permissible size.

The cyclitols would benefit from a more comprehensive introduction. A chapter should be included on the discovery of known cyclitols and of the enzymology producing them, as well as a chapter dedicated to the myo-inositol phosphates, the most studied cyclitols. In the current manuscript the link between cyclitols and human diseases (starting from chapter 2: diabetes; PICOS; hypertension; atherosclerosis; Alzheimer’s; carcinogenesis; allergy chapter 8) is just a list of useless information, often poorly referenced and/or incorrect. If the authors aim to discuss cyclitols in the context of human disease they should do so properly, starting with explaining the basic biology of these molecules, then integrating these facts with what is known of their tangible use in medicine. The authors review the beneficial use of fenugreek (Chapter 7: carcinogenesis) that may or may not contain D-pinitol. It certainly contains many other molecules, and linking the beneficial use of this plant just to D-pinitol is not science, it is pure speculation.

I added the data about concentration cyclitols in fenugreek. Lahuta, L.B.; Szablińska, J.; Ciak, M.; Górecki, R.J. The occurrence and accumulation of D-pinitol in fenugreek (Trigonella foenum graecumL.). Acta Physiol. Plant. 2018, 40, 155, doi: 10.1007/s11738-2734-4

The slim figure legends are another reason for concern; explaining the meaning of the figures would benefit the reader. For example, Figure 1 should state that cyclitols are not reducing sugars and explain the reason. Figure 2 shows that of the three methyl-derivates of myo-inositol, only one is synthesized from S-Adenosylmethionine with “IMT/catalysis”. From where the other two are synthesized is unclear. It is also not clear what IMT stands for: it is misspelled Massachusetts Institute of Technology or Inositol Methyl Transferase? Nowhere in the paper is IMT defined.

We corrected of Fig. I’ve added List of Abrreviations

The several tables are not referenced in the main text!  How this is possible? We added them into the text.

The two biggest jaw-dropping statements:

Line 63: “inositol triphosphate, a hormone regulating other hormones”

Inositol triphosphate is a cellular second messenger generated after the activation of phospholipase-C. This is a textbook example of signal transduction. Please read Chapter 16 - Essential Cell Biology. 4th Edition. by B. Alberts, D. Bray, K. Hopkin, A. Johnson, J. Lewis, M. Raff, K. Rober. Publisher: Garland Science. ISBN: 9780815344544.

I’ve added information: Myo-inositol is the precursor of inositol triphosphate, a second messenger generated after the activation of phospholipase-Cand regulating many hormones such as hyroid stimulating hormone (TSH), follicle stimulating hormone (FSH) and insulin [1,2].

1.    Chapter 16 - Essential Cell Biology. 4th Edition. by B. Alberts, D. Bray, K. Hopkin, A. Johnson, J. Lewis, M. Raff, K. Rober. Publisher: Garland Science. ISBN: 9780815344544.  

2.    M. Bizzarri, G. Carlomagno. Inositol: history of an effective therapy for Polycystic Ovary Syndrome.Eur Rev Med Pharmacol Sci 2014; 18 (13): 1896-1903

Line 374 “In mammals, the level of inositol in plasma is from 25 to 100 M” 

Myo-inositol’s solubility limit is ~1-2 M in water. How is it possible that in the much denser plasma its solubility increased by a factor of 100?

In mammals, the concentrations of inositol in plasma is 0.03 mM, while intracellular are several times higher than in the circulation, in cerebrospinal fluid 0.2 mM and the highest occurs in the brain - 6 mM [76]. Fisher et al., 2002. Journal of Neurochemistry, 82:736-754 

Misinformation:

Line 153 “Baquer et al. [37] used fenugreek seed powder orally (DP [D-pinitol] source) in rats”. 

Reference [37] does not state that fenugreek is a source of D-pinitol. Fenugreek as source of D-pinitol appears several times in the text, but there are no references indicating this plant as a source of D-pinitol. Bizarrely, the author’s own work, recently published in Acta Physiologiae Plantarum, is not cited.

 I’ve added an article - Lahuta, L.B.; Szablińska, J.; Ciak, M.; Górecki, R.J. The occurrence and accumulation of D-pinitol in fenugreek (Trigonella foenum graecumL.). Acta Physiol. Plant. 2018, 40, 155, doi: 10.1007/s11738-2734-4

In Table 2 is cited Davis et al., 2000 [94].

However, the manuscript list only 93 references. – I’ve corrected it.

Round  2

Reviewer 2 Report

The new version of the manuscript takes into account the majority of my early concerns. The English language, although not perfect, is substantially improved. There are not unclear statements. This review fairly discusses an important and interesting topic. The manuscript is now worth publication.